



# Long-term energy and $CO_2$ flux observations over an agricultural field in southeastern Tibetan Plateau

Anlun Xu[1, 2], Jian Li[3]

[1]Dali National Climate Observatory, Dali 671003, China
[2]Dali Mountain Meteorological Field Experiment Base, China Meteorological Administration, Dali 671003, China
[3]Chinese Academy of Meteorological Sciences, China Meteorological Administration, Beijing 100081, China

*Correspondence to*: Jian Li (lij@cma.gov.cn)

**Abstract.** Information on the exchange of energy, momentum and mass ($H_2O$, $CO_2$, $CH_4$, etc.) over complex topography is critical for determining the development of the boundary layer, carbon and water cycles, weather and climate. This

information can also improve the numerical modelling of physical atmosphere-land processes. Based on a 12-year (2007–2018) eddy covariance dataset over the Dali agricultural field in the southeastern Tibetan Plateau, we analysed the diurnal, seasonal and inter-annual changes in sensible heat flux ($H_s$), latent heat flux (LE) and $CO_2$ flux ($Fc$) and their meteorological controls on multiple timescales (half-hourly, daily, monthly, and yearly). The results show that both $H_s$ and LE have similar diurnal and seasonal variations, but the amplitude of LE is obviously larger than that of $H_s$ throughout the year, which

indicates that the LE plays a dominant role in surface heat exchange. The $Fc$ has a noticeable diurnal cycle, reaching its minimum around noon, and clear seasonal variations, reaching its minimum in the summer. The annual average $H_s$ increased from approximately 6 W $m^{-2}$ during 2007–2012 to 19 W $m^{-2}$ during 2013–2018, while the LE decreased from approximately 110 W $m^{-2}$ during 2007–2013 to 79 W $m^{-2}$ during 2014–2018. The Dali observational area is a carbon sink in all years, while the magnitude of net uptake decreases significantly from approximately 739 g C $m^{-2}$ $yr^{-1}$ during 2007–2013 to 218 g C

$m^{-2}$ $yr^{-1}$ during 2014–2018. The results also show that wind speed (WS) is the major control of $H_s$, while the product of WS and vapour pressure deficit (VPD) is the main driver of LE on different timescales. The net radiation ($R_n$) and soil temperature ($T_s$) have the largest effects on $Fc$ from the daily to monthly timescales, while the WS has the largest impact on annual total $Fc$.

## 1 Introduction

The Tibetan Plateau (TP), known as the "Roof of the World", stretches approximately 2500 km in the longitudinal direction and 1000 km in the latitudinal direction, and its average elevation exceeds 4 km above sea level (Fig. 1a), entering into the mid-level troposphere. Owing to its special atmosphere-land interactions, along with mechanical and thermodynamic effects, the TP exerts significant impacts on the evolution of atmospheric circulation, climate change and extreme weather events (Ye et al. 1957; Flohn 1957; Hahn and Manabe 1975; Liu and Chen 2000; Xu et al. 2008a). To understand the scientific

processes on the TP, such as atmosphere-land exchange processes and its weather and climate effects, a series of field





campaigns have been conducted over the TP and its surrounding areas since the 1970s (Duan et al. 2012; Li et al. 2015; Zhao et al. 2019). For instance, the first Qinghai-Xizang Plateau Meteorological Experiment (QXPMEX) was conducted from May to August 1979 (Tao et al. 1986), the second atmospheric Tibetan Plateau Field Experiment (TIPEX) was implemented in 1998 (Zhou et al. 2000) and a new integrated observational network was supported by the Japan International Cooperation

Agency Project (JICA/Tibet Project) during 2005–2009 (Xu et al. 2008b; Zhang et al. 2012). The Third Tibetan Plateau Atmospheric Scientific Experiment (TIPEX-III) was jointly initiated in 2013 and formally launched in 2014, with an 8–10 year implementation plan (Zhao et al. 2018, 2019). According to in situ observation data, many studies have reported the characteristics of surface water vapour, heat energy and $CO_2$ fluxes. However, the above field experiments were carried out in the main body of the TP, and most investigations focused on summer and were limited by observation parameters. There

is little information on the turbulent exchange over the southeastern TP.

The southeastern TP belongs to the transitional zone between the TP and Yunnan-Guizhou Plateau and is also located in a major water vapour path and a confluence zone between the South Asian monsoon and East Asian monsoon (Li et al. 2011). This area has complex topography with mountains, lakes, rivers, basins, meadows, forests, farmlands, wetlands, etc. Due to the unique location, complex terrain and a lack of in situ observation data in this area, the results of some atmospheric

general circulation models show obvious deviations (Yu et al. 2000). Moreover, basic information on the land surface and atmosphere in this region plays an important role in the evolution of atmospheric circulation over East Asia as well as weather and climate change in the Yangtze River basin and southwestern China (Hua et al. 2008; Liu et al. 2009). Therefore, revealing the characteristics of the local weather and climate in this region is of great importance, and quantification of atmosphere-land interaction parameters will improve the numerical models and parameterization schemes of the atmospheric

boundary layer in southwest China and even the East Asia region.

Under the support of the JICA/Tibet Project (Xu et al. 2008b; Zhang et al. 2012), we built a planetary boundary layer (PBL) site (Fig. 1c) in December 2006 to continuously observe the micrometeorology and energy exchange. The aims of this study are to characterize the diurnal, seasonal and inter-annual variations in energy and $CO_2$ fluxes and their meteorological controlling factors over the southeast extension of the TP.

## 55   2 Materials and methods

### 2.1 Study site description

The PBL site (100°10′36″ E, 25°42′28″ N, 1977.7 m above sea level) lies in Dali Bai Autonomous Prefecture, Yunnan Province, China. The site is surrounded by open and flat farmland (Fig. 1b-c), where broad beans are mainly planted in the dry season (November to April of the next year) and rice is mainly planted in the wet season (May to October). The

maximum broad bean height at peak growth can reach up to 1.0 m, and the rice height can reach up to 1.2 m. The climate at the measurement site features a monsoon climate of the low latitude plateau. Based on the observational data from 1951 to 2018 at an automatic weather station (AWS) located approximately 220 m southwest of the study site, the annual average air



temperature is 15.1 °C, and the annual total precipitation is 1053.4 mm, with nearly 85% of precipitation falling in the wet season. The annual average wind speed is 2.4 m s$^{-1}$, and the strongest wind speed is 40.8 m s$^{-1}$, with gale commonly

experienced from the winter season through the spring season. There are two types of prevailing winds with east-southeasterly winds in the daytime and west-northwesterly winds occurring at night.

### 2.2 Measurement setup

A 20-m tower was set up to acquire air temperature ($T_a$), relative humidity (RH), wind speed (WS) and wind direction (WD) profiles along with sensible heat flux ($H_s$), latent heat flux (LE), and $CO_2$ flux ($Fc$) in the near-surface layer. Sensors

recording $T_a$, RH (HMP45C, Vaisala), WS and WD (034B, Met One) were mounted at heights of 2, 4, 10, and 20 m on the tower. $T_a$, LE, and $Fc$ were directly determined by an eddy covariance (EC) system containing a three-dimensional sonic anemometer (CSAT3, Campbell) and an open-path $CO_2/H_2O$ infrared gas analyser (LI-7500, LI-COR). The distance between the two sensor heads was 18 cm. Both instruments were operated at a height of 5 m on the tower with a sampling frequency of 10 Hz. Net radiation flux ($R_n$) was measured at 1.5 m above the surface with a four-component net radiometer

(CNR1, Kipp & Zonen). Soil temperature ($T_s$) and soil water content (SWC) at depths of 4, 10, 20, 60, and 100 cm were measured with temperature probes (Model 107, Campbell) and water content reflectometers (CS616, Campbell), respectively. Soil heat flux ($G_s$) at depths of 4, 10, and 20 cm was measured by soil heat flux plates (HFP01, Hukseflux). High-frequency 10-Hz raw data were gathered using a data logger (CR3000, Campbell) with a 1 GB CF card, and low-frequency 10-min raw data were collected using a data logger with a 64 MB CF card.

### 2.3 Data processing

To obtain 30-min eddy covariance flux data, the 10-Hz raw data were done using EddyPro (version 6.2.1, LI-COR). The processing steps used here included spike removal (Vickers and Mahrt 1997), double rotation for tilt correction (Kaimal and Finnigan 1994), spectral corrections (Moncrieff et al. 1997, 2004) and Webb-Pearman-Leuning (WPL) corrections (Webb et al. 1980). In addition, a quality check of the 30-min fluxes was performed using the steps proposed by Foken et al. (2004),

including the steady state test and the well-developed turbulence test. Gap filling fluxes were performed with the REddyPro package (https://www.bgc-jena.mpg.de/bgi/index.php/Services/REddyProcWebRPackage) in the cross-platform language R (Wutzler et al. 2018).

### 3 Results

### 3.1 Meteorological conditions

The meteorological parameters, such as daily integrated global solar radiation ($R_g$), daily average $T_a$, RH, WS, $T_s$, SWC and daily total precipitation (PPT), all display a clear dry and wet seasonal variation (Fig. 2). The daily integrated $R_g$ ranges from





0.66 to 31.57 MJ m$^{-2}$ d$^{-1}$, with a 12-year average value of 17.35 MJ m$^{-2}$. The annual integrated $R_g$ ranges from 5066.1 to 6702.7 MJ m$^{-2}$ yr$^{-1}$, with an average value of 6154.1 MJ m$^{-2}$ and a standard deviation of 450.2 MJ m$^{-2}$ (Table 1).

The $T_a$ at different heights of the near-surface layer in the wet season are higher than those in the dry season. For monthly average $T_a$, the fastest increase period occurs between March and April, whereas a decrease period occurs between October and November. The daily average $T_a$ values at heights of 2, 4, 10, and 20 m range from 0.4 to 24.7, 0.6 to 24.3, 0.3 to 24.2, and 0.4 to 24.4 °C, with 12-year average values of 16.1, 16.1, 16.2, and 16.5 °C, respectively. The maximum and minimum daily average $T_a$ values are recorded in June 2014 and December 2013.

The RH of the near-surface layer usually decreases with measurement height, and in the wet season, it is higher than in the dry season. The monthly average RH reaches the highest in August and the lowest in February. The daily average RH values at heights of 2, 4, 10, and 20 m range from 14.5 % to 100 %, 13.4 % to 99.5 %, 12.4 % to 95.4 %, and 11.8 % to 96.3 %, with 12-year average values of 64.6 %, 62.3 %, 57.2 %, and 56.8 %.

The WS of the near-surface layer commonly increases with measurement height, and in the wet season, it is weaker than that in the dry season. The monthly average WS reaches the strongest in February and the weakest in September. The maximum daily average WS values at heights of 2, 4, 10, and 20 m are observed in January 2008 (7.1, 8.4, 9.9, and 11.1 m s$^{-1}$). The 12-year average WS values at heights of 2, 4, 10, and 20 m are 1.4, 1.7, 2.4, and 2.8 m s$^{-1}$, respectively.

Local complicated terrain often affects air flow within the atmospheric boundary layer, which not only changes wind speed but also changes wind direction. In the study area, the predominant wind direction is easterly and east-southeasterly throughout the year (Fig. 3). The prevailing and sub-prevailing winds at a height of 10-m display a clear diurnal cycle, with east-southeasterly and easterly winds during the daytime and westerly and static winds at night (Fig. 4). The time when the prevailing winds switch exhibits good correspondence with the sunrise and sunset (solar radiation heating effect). This result may be chiefly caused by the shapes of the Diancangshan Mountains and Erhai Lake, which extends from northwest to southeast.

The $T_s$ values at different soil depths in the wet season are higher than those in the dry season. The monthly average $T_s$ reaches the highest in July and the lowest in January. The $T_s$ values in the shallow soil layer are higher than those in the deep soil layer in the wet season; conversely, $T_s$ in the shallow soil layer is lower than that in the deep soil layer in the wet season. The daily average $T_s$ values at depths of 4, 10, 20, 60, and 100 cm range from 4.2 to 27.3, 4.9 to 25.3, 7.1 to 26.8, 10.1 to 23.4, and 11.5 to 21.9 °C, with 12-year average values of 16.4, 16.1, 16.5, 16.6, and 16.6 °C.

The maximum daily total PPT value is observed in October 2015 (121.0 mm d$^{-1}$), and the maximum monthly total PPT is observed in June 2008 (336.7 mm month$^{-1}$). The annual total PPT fluctuates from 732.9 mm yr$^{-1}$ (2013) to 1322.8 mm yr$^{-1}$ (2008), with an average value of 962.1 mm and a standard deviation of 179.3 mm (Table 1). The PPT during the wet season ranges from 643.2 to 1171.9 mm, accounting for 75.9 % to 93.6 % of the annual total PPT. Because of the frequent drought influence, the PPT is below the 12-year average value for five consecutive years from 2010 to 2014.

The SWC values at different depths in the wet season are larger than those in the dry season, and their values also respond significantly to PPT. With the start of the rainy season (commonly occurring from late May to early June), the SWC rapidly





increases and reaches its maximum. The daily average SWC values at depths of 4, 10, 20, 60, and 100 cm range from 0.07 to 0.73, 0.08 to 0.70, 0.19 to 0.59, 0.27 to 0.48, and 0.35 to 0.46 $m^{-3}\, m^{-3}$, with 12-year average values of 0.37, 0.38, 0.42, 0.40, and 0.42 $m^{-3}\, m^{-3}$, respectively.

### 3.2 Diurnal, seasonal and inter-annual variations in energy fluxes

The monthly average diurnal variations in the energy balance components from 2007 to 2018 are shown in Fig. 5a. For each month, the $R_n$, $H_s$, LE and $G_s$ all have clear, similar diurnal courses. The $R_n$ remains positive during the day, reaching its peak at approximately 13:30, and negative values occur at night, reaching its valley at approximately 20:30. The half-hourly $R_n$ values range from –100 to 1076 W $m^{-2}$. The monthly valleys and peaks of $R_n$ vary from –67 to –32 W $m^{-2}$ and from 418 to 513 W $m^{-2}$, respectively. The $H_s$ value gradually increase after sunrise, becoming positive at approximately 09:00, and reach

a maximum at approximately 13:30. Then, the $H_s$ values decrease and become negative after sunset, reaching a minimum of approximately 20:30. The monthly maximal and minimal $H_s$ values range from 66 to 127 W $m^{-2}$ and from –52 to –7 W $m^{-2}$, respectively. The LE remains positive throughout the day and reaches its maximum at approximately 14:00. The monthly maximal and minimal LE values range from 205 to 337 W $m^{-2}$ and 3 to 20 W $m^{-2}$, respectively. The diurnal average $G_s$ is very low throughout the year, with the valley occurring in November (–25 W $m^{-2}$) and the peak occurring in June (47 W $m^{-2}$).


The seasonal and inter-annual variations in the daily and monthly average energy balance components from 2007 to 2018 are shown in Fig. 6a-d and Fig. 7a-d, respectively. For each year, the $R_n$, $H_s$, LE and $G_s$ values are all larger in the wet season than in the dry season. The daily and monthly average $R_n$ values vary from –88 to 1005 W $m^{-2}$ and –59 to 365 W $m^{-2}$, with a 12-year average value of 112.8 W $m^{-2}$. The daily and monthly average $H_s$ values range from –38 to 73 W $m^{-2}$ and –25 to 34

W $m^{-2}$, with a 12-year average value of 12.7 W $m^{-2}$. The daily and monthly average LE ranges from 10 to 255 W $m^{-2}$ and 41 to 161 W $m^{-2}$, with a 12-year average value of 96.8 W $m^{-2}$. The daily and monthly average $G_s$ values range from –24 to 26 W $m^{-2}$ and –7 to 8 W $m^{-2}$, with a 12-year average value of –0.4 W $m^{-2}$.

### 3.3 Diurnal, seasonal and inter-annual variations in $CO_2$ flux

The monthly average diurnal variations in $F_c$ from 2007 to 2018 are shown in Fig. 5b. For each month, the $F_c$ has a

noticeable diurnal cycle with positive values at night reaching their maximum at approximately 07:00 and negative values during the day reaching their minimum at approximately 13:00. This finding indicates that the study area acts as a weak carbon source at night and a carbon sink during the day. The monthly maximal $CO_2$ release and uptake rates range from 1.46 to 4.54 $\mu$mol $m^{-2}\, s^{-1}$ and from 2.59 to 16.62 $\mu$mol $m^{-2}\, s^{-1}$.

The seasonal and inter-annual variations in the daily and monthly average $F_c$ values from 2007 to 2018 are shown in Fig.

6e and Fig. 7e, respectively. For each year, the $F_c$ is higher during the transitional periods (between April and May, between October and November) than that at the other times. The daily and monthly average $F_c$ values range from –11.59 to 4.13


µmol m$^{-2}$ s$^{-1}$ and –7.79 to 1.67 µmol m$^{-2}$ s$^{-1}$. The annual total $F$c varies widely from –966.9 to –75.6 g C m$^{-2}$ yr$^{-1}$, with an average value of -522.0 g C m$^{-2}$ and a standard deviation of 295.7 g C m$^{-2}$ (Table 1).

## 4 Discussion

### 4.1 Controlling factors in $H_s$

The correlation matrices of $H_s$ and meteorological factors over different temporal scales are shown in Fig. 8–11. Compared to other meteorological factors, the correlation coefficients between $H_s$ and $R_n$, WS, as well as the product of WS and VPD are highest on half-hourly scales, with values of 0.47, –0.35, and –0.35. On daily scales, the correlation coefficients between $H_s$ and WS, $R_n$, as well as the product of WS and ΔT are highest, with values of –0.56, 0.53, and –0.33. On monthly scales, the correlation coefficients between $H_s$ and WS as well as the product of WS and ΔT, WS and VPD are highest, with values of –0.75, –0.48, and –0.58. On yearly scales, the correlation coefficients between $H_s$ and WS, $R_n$, as well as the product of WS and VPD are highest, with values of –0.96, –0.39, and –0.65. These results mean that WS is the main meteorological factor controlling $H_s$ on different timescales.

### 4.2 Controlling factors in LE

The correlation matrices of LE and its environmental controls over different temporal scales are shown in Fig. 8–11. Relative to other environmental variables, the correlation coefficients between LE and VPD, $R_n$, as well as the product of WS and VPD are highest on half-hourly scales, with values of 0.43, –0.42, and –0.41. On daily scales, the correlation coefficients between LE and $G_s$, $R_n$, $T_a$, $T_s$ as well as VPD are highest, with values of 0.62, 0.61, 0.54, 0.43, and 0.41. On monthly scales, the correlation coefficients between LE and $G_s$, $T_a$, $R_n$, VPD as well as the product of WS and VPD are highest, with values of 0.71, 0.57, 0.51, 0.43, and 0.46. On yearly scales, the correlation coefficients between LE and $F$c, WS, as well as the product of WS and VPD are highest, with values of –0.91, 0.88, and 0.57. The above results indicate that the product of WS and VPD is the main environmental variable controlling LE on different timescales.

### 4.3 Controlling factors in $CO_2$ flux

The correlation matrices of $F$c and its drivers over different temporal scales are shown in Fig. 8–11. On half-hourly scales, the correlation coefficients between $F$c and its drivers are relatively low. The highest correlation coefficient is found for the relationships between $F$c and both $R_n$ and $G_s$ (with a value of –0.14). Compared to other drivers, the correlation coefficients between $F$c and $R_n$, $T_s$ as well as RH are highest on the daily scales with values of –0.41, –0.33, and 0.30. On monthly scales, the correlation coefficients between $F$c and $T_s$, $R_n$, $T_a$, RH, PPT as well as ΔT are highest, with values of –0.40, –0.38, –0.35, –0.34, –0.32, and –0.33. On yearly scales, the correlation coefficients between $F$c and WS, $G_s$ $R_n$, SWC, as well as the product of WS and VPD are highest, with values of –0.92, –0.53, 0.40, –0.40, and –0.64.



**5 Conclusions**

In this study, we report the variations in energy and $CO_2$ fluxes and their meteorological drivers at various temporal scales from 2007 to 2018 at the Dali observational site over the southeast extension of the Tibetan Plateau. The $R_n$, $H_s$, LE and $G_s$ all have similar diurnal courses, reaching their maximum values around noon and attaining their minimum values around early evening. Moreover, these factors present obvious seasonal changes with larger values in the wet season than in the dry season. The averages and standard deviations of $R_n$, $H_s$, LE and $G_s$ for 12 years are 112.8±12.9, 12.7±7.1, 96.8±16.4, and –0.4±0.5 W m$^{-2}$, respectively. The $F_c$ has a noticeable diurnal cycle with positive values at night and negative values during the daytime, and it also exhibits clear seasonal changes with the highest values during the transitional periods between the dry and wet seasons. The annual total $F_c$ values fluctuate from –966.9 to –75.6 g C m$^{-2}$ yr$^{-1}$, with an average value of –522.0 g C m$^{-2}$. This result suggests that the study area acts as a weak carbon source at night and a carbon sink during the day, whereas it acts as an overall carbon sink over all years. Each meteorological driver has a distinct effect on $H_s$, LE, and $F_c$ at different temporal scales. WS shows a decreasing trend according to the correlation coefficients for $H_s$, while the product of WS and VPD made a high contribution to LE from the half-hourly to yearly timescales. For $F_c$, both $R_n$ and $T_s$ have high contributions at the daily to monthly timescales, but the WS is the most important at the yearly timescale.

**Competing interest**

The authors declare that they have no conflict of interest.

**ACKNOWLEDGEMENTS**

This study was supported by the National Natural Science Foundation of China (Nos. 91637210, 41875123, and 91737306) and Jiangsu Collaborative Innovation Center for Climate Change.

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





**Table 1: Annual average wind speed (WS, m s$^{-1}$) at a height of 10 m, air temperature ($T_a$, °C), relative humidity (RH, %) and vapour pressure deficit (VPD, kPa) at a height of 4 m, soil temperature ($T_s$, °C) and soil water content (SWC, m$^{-3}$ m$^{-3}$) at a soil depth of 4 cm, net radiation ($R_n$, W m$^{-2}$), albedo, sensible heat flux ($H_s$, W m$^{-2}$), latent heat flux (LE, W m$^{-2}$), Bowen ratio (β), and**
**soil heat flux ($G_s$, W m$^{-2}$) at a soil depth of 4 cm; and annual total global solar radiation ($R_g$, MJ m$^{-2}$), CO$_2$ flux ($Fc$, g C m$^{-2}$), precipitation (PPT, mm), evapotranspiration (E, mm) at the Dali site from 2007 to 2018**

| Year | WS | $T_a$ | RH | VPD | $T_s$ | SWC | $R_n$ | Albedo | $H_s$ | LE | β | $G_s$ | $R_g$ | $F_c$ | PPT | E |
|---|---|---|---|---|---|---|---|---|---|---|---|---|---|---|---|---|
| 2007 | 2.6 | 16.6 | 68.0 | 0.66 | 18.0 | 0.320 | 113.3 | 0.180 | 3.4 | 106.2 | 0.03 | −0.1 | 5066.1 | −625.9 | 1139.8 | 2847.9 |
| 2008 | 2.5 | 15.8 | 68.3 | 0.62 | 18.4 | 0.381 | 146.7 | 0.183 | 7.6 | 109.0 | 0.07 | 0.1 | 5659.3 | −792.3 | 1322.8 | 2988.3 |
| 2009 | 2.7 | 16.4 | 62.1 | 0.76 | 17.3 | 0.352 | 114.8 | 0.169 | 7.4 | 114.5 | 0.06 | 0.4 | 6702.7 | −737.1 | 1031.2 | 3090.7 |
| 2010 | 2.8 | 16.3 | 62.6 | 0.74 | 16.4 | 0.394 | 128.1 | 0.187 | 3.3 | 113.8 | 0.03 | −0.1 | 6043.6 | −966.9 | 863.8 | 3169.8 |
| 2011 | 2.5 | 15.7 | 62.4 | 0.72 | 15.9 | 0.426 | 102.7 | 0.199 | 9.2 | 109.5 | 0.08 | −0.4 | 6126.7 | −788.3 | 740.1 | 3028.3 |
| 2012 | 2.7 | 16.4 | 57.7 | 0.83 | 15.8 | 0.422 | 101.4 | 0.215 | 6.8 | 113.4 | 0.06 | −0.5 | 6120.7 | −634.5 | 853.2 | 3238.1 |
| 2013 | 2.4 | 15.9 | 57.6 | 0.84 | 16.1 | 0.423 | 104.1 | 0.201 | 15.9 | 100.5 | 0.16 | −0.1 | 6154.7 | −627.0 | 732.9 | 2788.3 |
| 2014 | 2.4 | 16.5 | 58.0 | 0.85 | 15.9 | 0.332 | 109.4 | 0.194 | 17.1 | 84.6 | 0.20 | −0.2 | 6687.1 | −305.2 | 812.9 | 2339.9 |
| 2015 | 2.0 | 16.1 | 61.0 | 0.81 | 15.9 | 0.326 | 102.7 | 0.186 | 18.1 | 81.5 | 0.22 | −1.1 | 6440.7 | −412.8 | 984.9 | 2276.8 |
| 2016 | 1.9 | 15.9 | 64.8 | 0.70 | 15.3 | 0.374 | 109.5 | 0.165 | 19.8 | 75.4 | 0.26 | −1.2 | 6069.8 | −187.6 | 1153.6 | 2184.9 |
| 2017 | 2.0 | 16.1 | 63.6 | 0.73 | 16.4 | 0.381 | 110.1 | 0.176 | 20.6 | 79.5 | 0.26 | −0.4 | 6356.9 | −75.6 | 909.8 | 2297.2 |
| 2018 | 1.9 | 15.5 | 63.0 | 0.72 | 16.9 | 0.344 | 110.5 | 0.187 | 23.0 | 73.7 | 0.31 | −0.5 | 6421.6 | −110.9 | 1000.7 | 2162.2 |
| Average | 2.4 | 16.1 | 62.4 | 0.75 | 16.5 | 0.373 | 112.8 | 0.187 | 12.7 | 96.8 | 0.15 | −0.4 | 6154.1 | −522.0 | 962.1 | 2701.0 |
| SD | 0.3 | 0.3 | 3.6 | 0.07 | 0.9 | 0.038 | 12.9 | 0.014 | 7.1 | 16.4 | 0.10 | 0.5 | 450.2 | 295.7 | 179.3 | 416.6 |





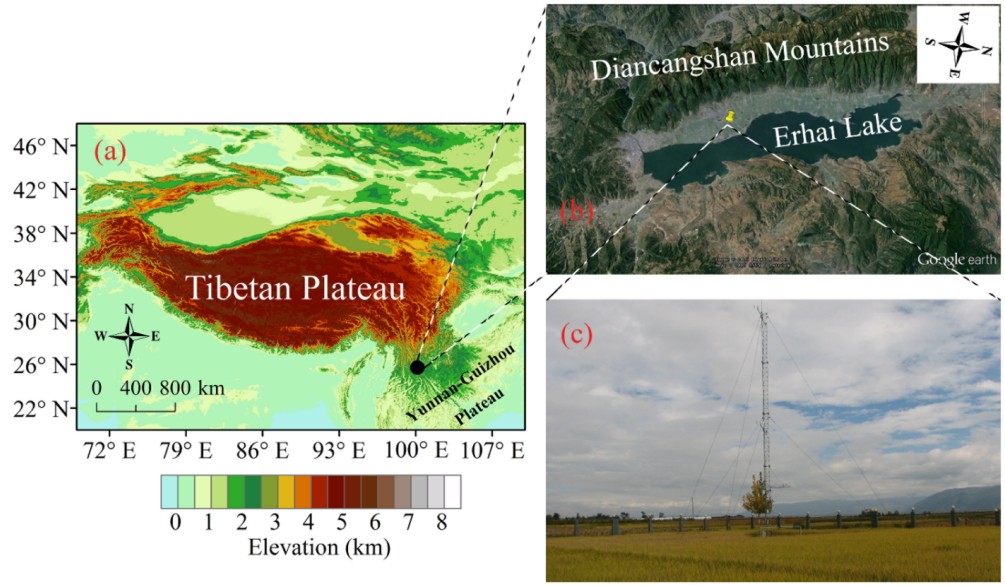


Figure 1: Topographic map of the Tibetan Plateau (a), location of the study area (map from © Google Earth) (b) and picture of the Dali planetary boundary layer site (c).



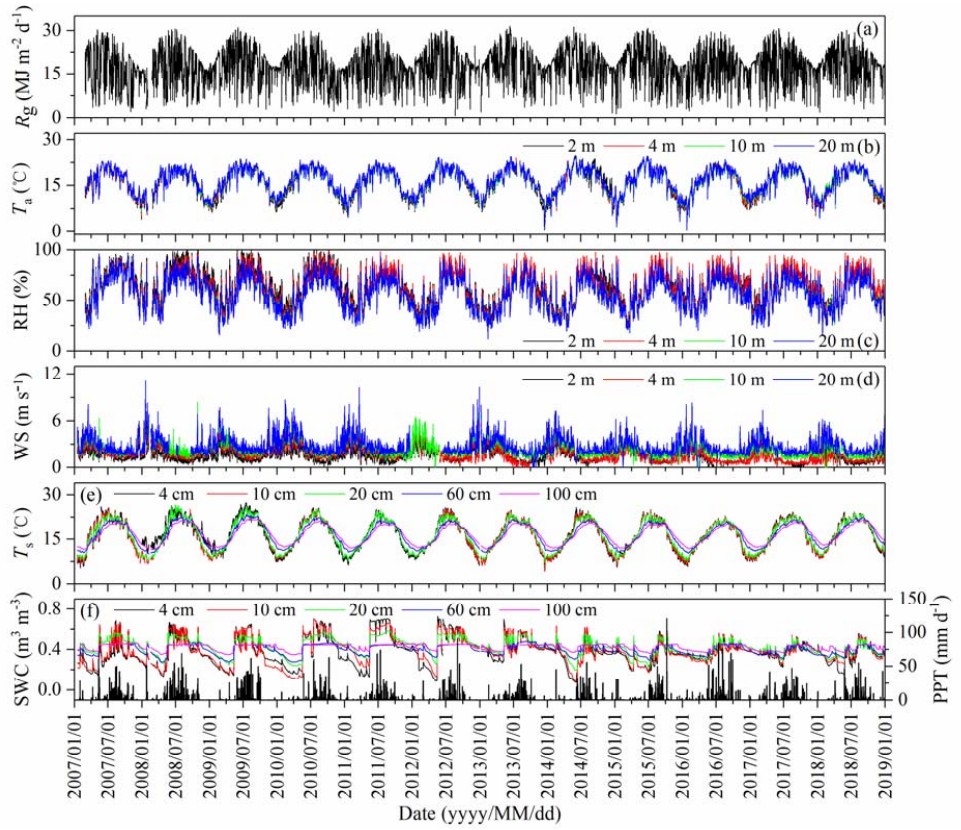

**Figure 2: Daily integrated global solar radiation ($R_g$), daily average air temperature ($T_a$), relative humidity (RH), wind speed (WS), soil temperature ($T_s$), soil water content (SWC) and daily total precipitation (PPT) from 2007 to 2018.**





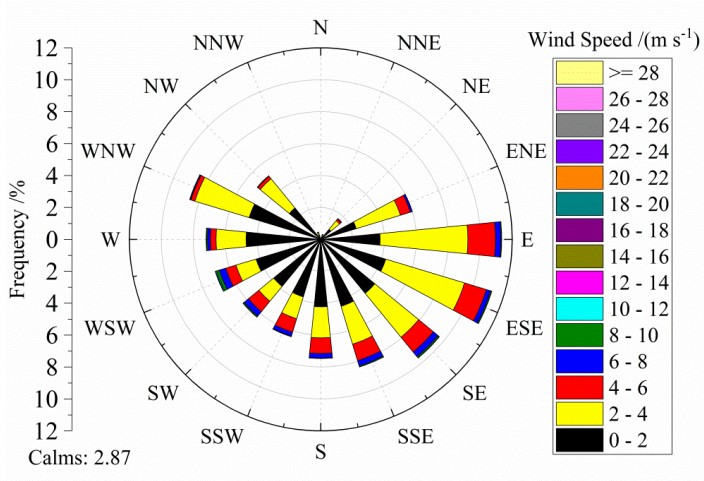

**Figure 3: Wind rose at a height of 10 m from 2007 to 2018.**

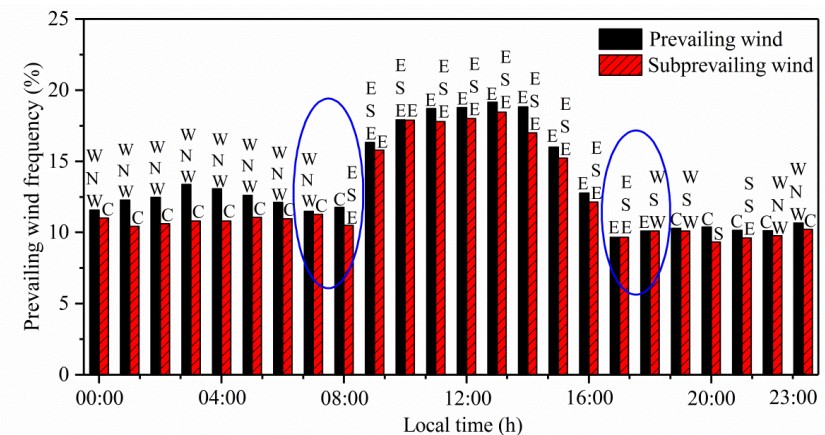

**Figure 4: Diurnal variations in prevailing wind frequency at a height of 10 m from 2007 to 2018. Ellipses indicate the times of sunrise and sunset.**

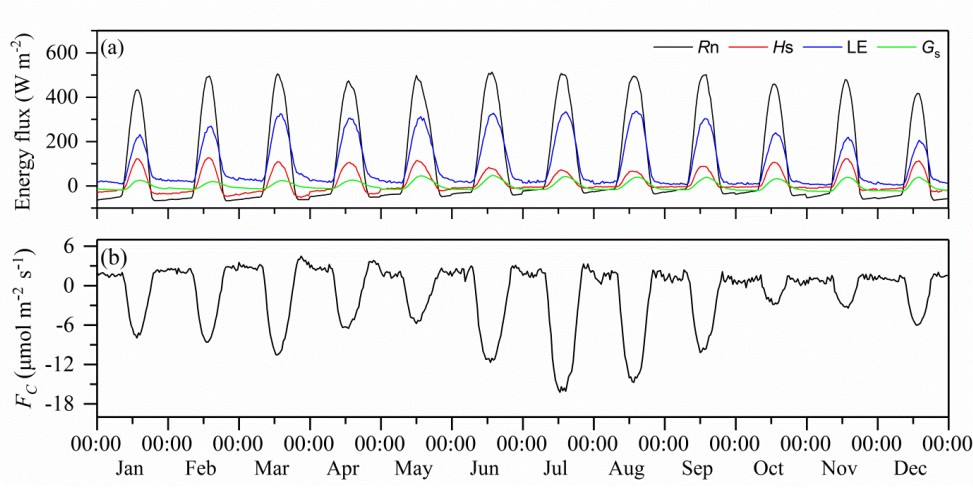

**Figure 5: Monthly average diurnal variations in energy balance components, including net radiation ($R_n$), sensible heat flux ($H_s$), latent heat flux (LE), soil heat flux ($G_s$), and $CO_2$ flux (Fc) from 2007 to 2018.**





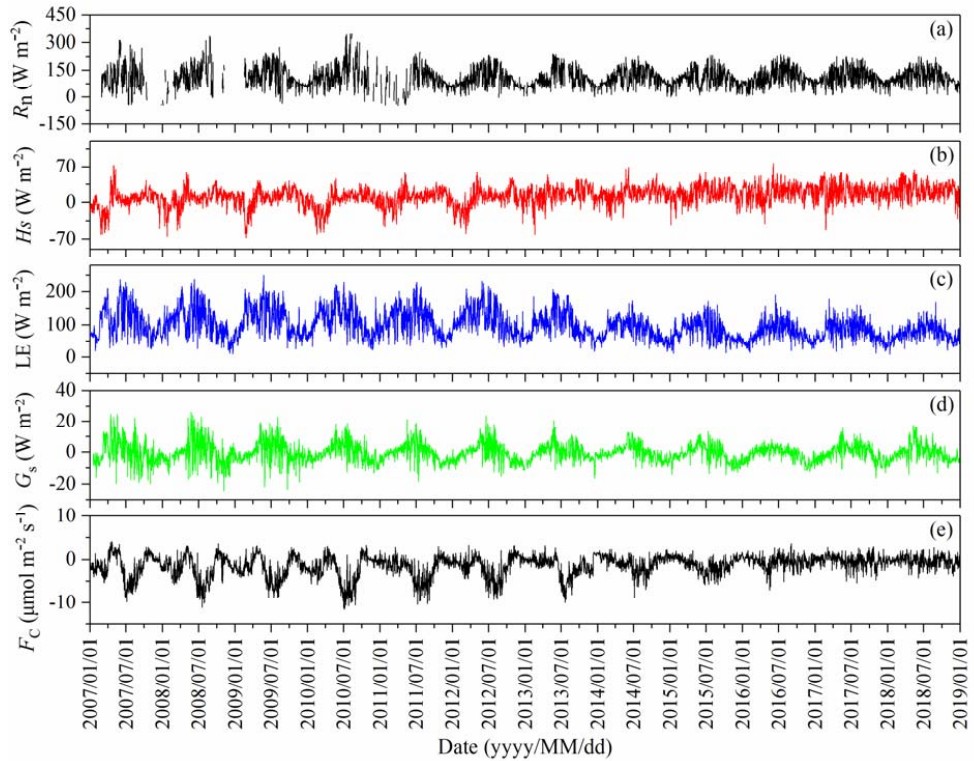


**Figure 6: Seasonal and inter-annual variations in daily average energy balance components, including net radiation ($R_n$), sensible heat flux ($H_s$), latent heat flux (LE), soil heat flux ($G_s$), and $CO_2$ flux ($F$c) from 2007 to 2018.**



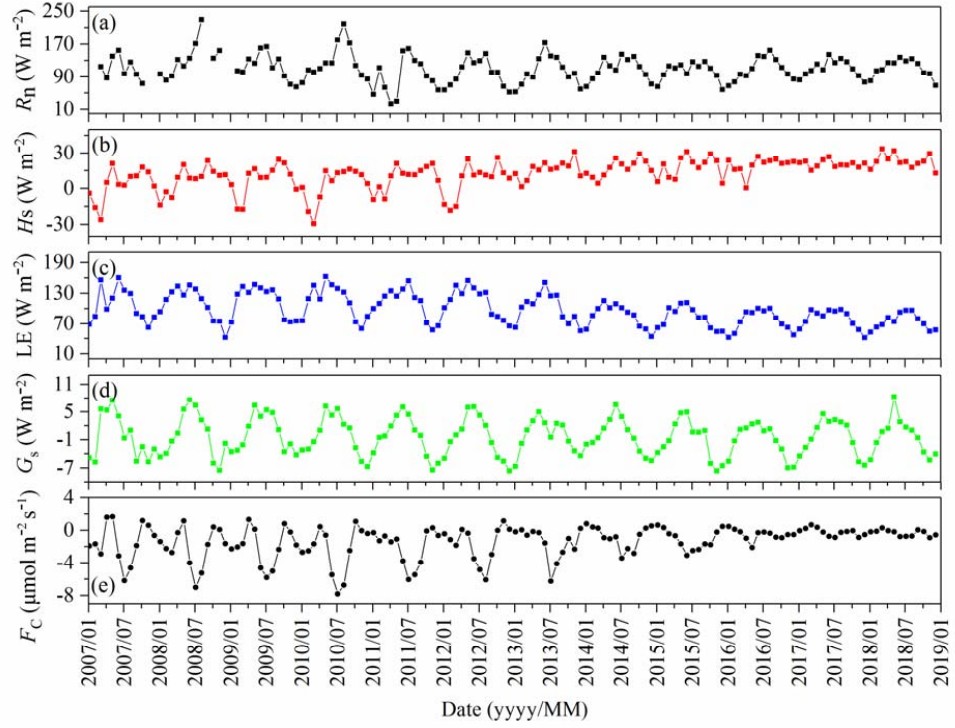

**Figure 7: Seasonal and inter-annual variations in monthly average energy balance components, including net radiation ($R_n$), sensible heat flux ($H_s$), latent heat flux (LE), soil heat flux ($G_s$), and $CO_2$ flux ($F_c$) from 2007 to 2018.**





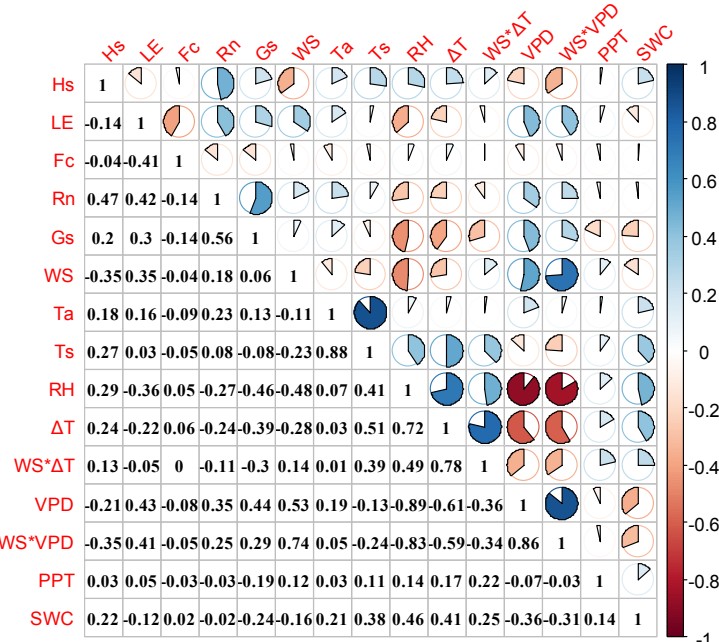

**Figure 8: Correlation matrix of turbulent fluxes including half-hourly sensible heat ($H_s$), latent heat (LE) and $CO_2$ flux ($Fc$) and meteorological factors including half-hourly net radiation ($R_n$), wind speed (WS), air temperature ($T_a$), relative humidity (RH), water vapour pressure deficit (VPD) at a height of 4 m, temperature difference between the soil surface and atmosphere (ΔT), the**

**product of WS and ΔT (WS\*ΔT), the product of WS and VPD (WS\*VPD), soil temperature ($T_s$), soil heat flux ($G_s$) and soil water content (SWC) at a soil depth of 4 cm, and precipitation (PPT) from 2007 to 2018.**



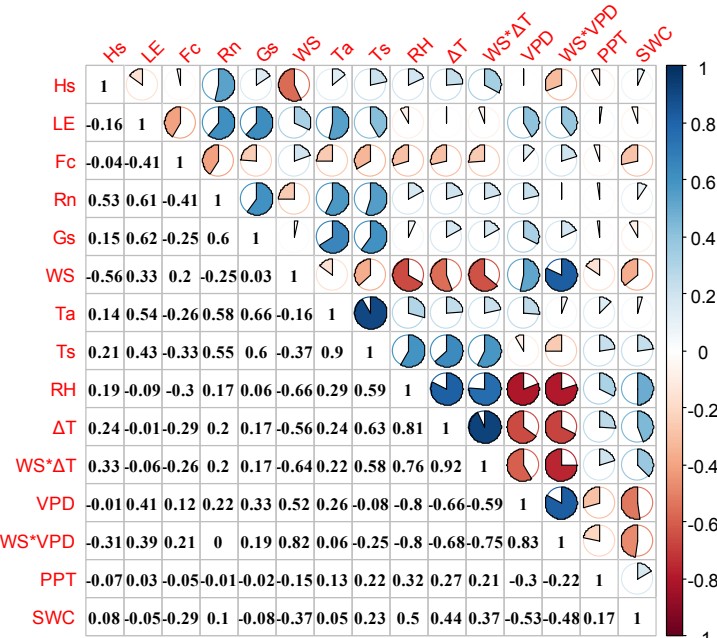

**Figure 9: Same as Figure 8, but for the daily average values of turbulent fluxes and meteorological factors other than precipitation, while precipitation is the daily total value.**





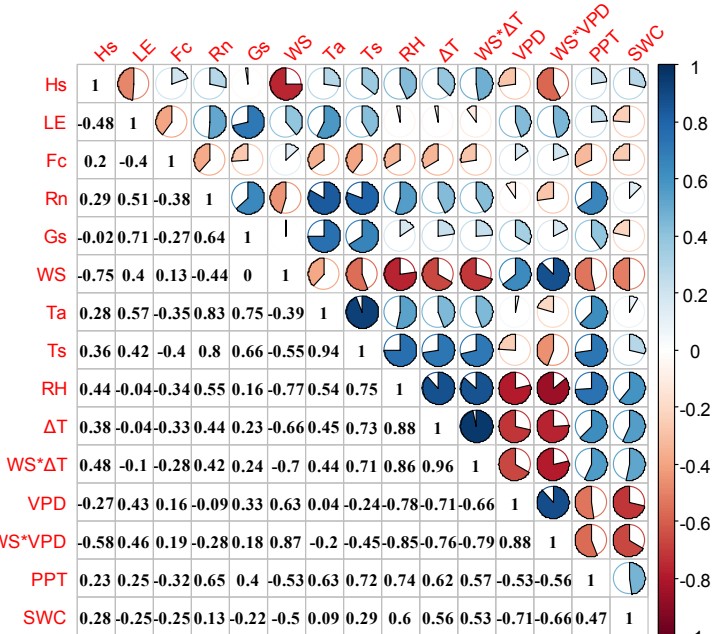


**Figure 10: Same as Figure 8, but for the monthly average values of turbulent fluxes and meteorological factors other than precipitation, while precipitation is the monthly total value.**



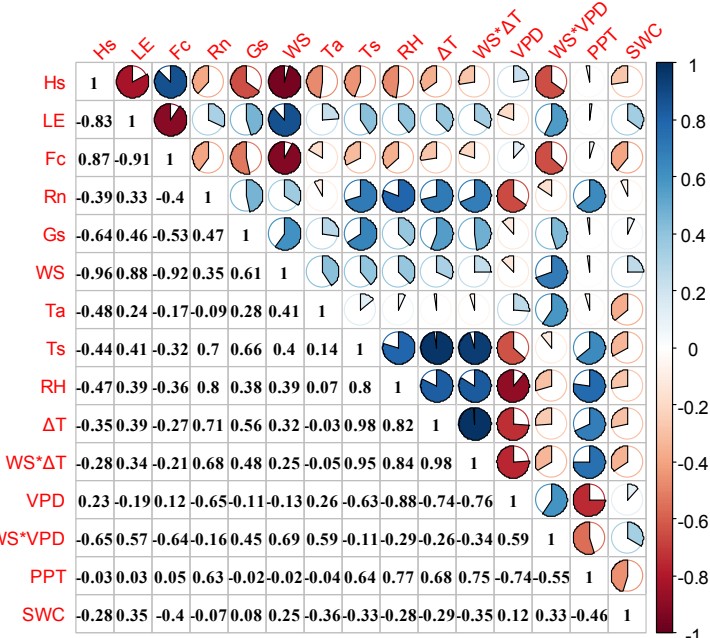

**Figure 11: Same as Figure 8, but for the annual average values of turbulent fluxes and meteorological factors other than**

**precipitation and CO$_2$ flux, while precipitation and CO$_2$ flux are the annual total values.**