# Peer review of "Long-term energy and CO2 flux observations over an agricultural field in southeastern Tibetan Plateau"

_Atmospheric Chemistry and Physics, 2019_

## Referee Comment (RC1) · Anonymous Referee #1 · 9 Mar 2020

General comments: This study presents more than a decadal continuous flux measurement in an agricultural field in southeastern Tibetan Plateau, which is very good data source to investigate the characteristics of the atmosphere-land interactions and turbulent exchanges of energy and mass in that region. However, after I read through the manuscript several times, I did not get more interesting findings except for knowing some numbers and magnitudes of several fluxes, which I think the study lacks significance and scientific contribution and therefore may need substantial work and more in-depth analyses and discussion. As the authors described in the introduction, I agree it is very important to understand the atmosphere-land interactions in the transitional zone between the TP and Yunnan-Guizhou Plateau. However, the study site is an open

and flat agricultural field and I wonder how well the flux measurements at the specific study site would be representative to the air-land interaction parameters in the transitional region, where more heterogeneity in land cover or topography may exist. The authors may need to stress more about the unique aspect of their study and reconsider: are the findings/results/data really relevant to what is currently described in the introduction and objectives? I also found that most of the results section is fully packed with numbers rather than result interpreting, for example, the authors spent one full page reporting all the meteorological data which are already clearly shown in Figure 2. I would suggest the authors could present the results by showing the key information in a concise way and including more in-depth analysis. In addition, all the discussion is about the correlation analysis results, which should be moved to the results section. Therefore, I think the manuscript may need a better organization overall. I also noticed that as one of the key results, there is an obvious step change in LE, H, and Fc after 2014 (Figure 6 or 7). In particular, Hs becomes very noisy and doesn't show any obvious seasonal pattern after 2014, which also applies to the seasonal variability of Fc. This is very interesting, but I wonder whether it is really caused by the environmental controls or just a systematic error. Unfortunately, the authors could not provide any explanation and discussion on this specific pattern of the inter-annual variability.

Specific comments: Line 30-37: what kind of field campaigns are they? Are these studies all about turbulent exchanges and relevant to this study? Line 56: The study site is a cropland. However, from the google map (Figure1b), it seems the site is very close to the Erhai lake. Without a scale on the map, it is difficult to figure out whether there are any signals coming from the adjacent non-cropland land cover (e.g. the lake). Line 90: Where are the global radiation and precipitation measurements coming from? Please describe in the methods section. Line 90-106: This part of results is literally repeating the same information shown in Figure 2. Suggest rewriting the results in a concise way that shows the key information rather than reporting the numbers. Line 111-113: suggest moving this sentence to discussion. Line 117-118: again, reporting numbers here. See my comment above. Line 131-140: one could easily get this

information from the figure. Line 170-185: these correlation analysis results are still "results", which should not be presented in the discussion. Suggest moving it the results section. Line 195: the inter-annual variability of annual carbon sink is really large, but I couldn't find there are any drivers or controls on this strong variability. Line 198: are there any physical or biophysical explanation on why wind speed affected the Hs? Figure1b: please add the scale to the map and rotate the map with N facing upward. Figure3: how to distinguish the wind >=28 m/s and 2-4 m/s, both are in yellow. Figure5: suggest adding the inter-annual variability of the monthly averaged diurnal patterns by showing the error bars. Figure 6 and 7 basically show same information. Suggest deleting one.

Technical corrections: Line 71: Hs instead of Ta? Line 81: suggest replacing "done" with "processed" Line 82: suggest deleting "used here" Line 92: round up 5066.1 to 5066. And please be careful about the significant digits throughout the manuscript.

---

## Referee Comment (RC2) · Anonymous Referee #2 · 18 Mar 2020

The manuscript provides several analisys regarding the behaviour and main controlling factors of H. LE and CO2 fluxes using more than 10 years. Despite the very relevant and valuable database, the manuscrip does not provide any relevant results beyond the analysis of relationships between variables and a visual and statistical description of the behavior of flows at different time scales. Additionally, regarding the structure of the manuscript, the results section mainly shows several "numbers" that are not easy to follow and the discussion section does not provide any relevant explanation to the behave of the fluxes or their main factors.

[Figure]

2020.

---

## Author Comment (AC1) · 11 May 2020

Dear Editor,

We thank the two anonymous reviewers for their invaluable and constructive comments to improve the manuscript. Our point-by-point response to the review comments are listed below as shown in black. Any page and line numbers in our response refer to the revised manuscript.

**Anonymous Referee #1**

**General comments:**

This study presents more than a decadal continuous flux measurement in an agricultural field in southeastern Tibetan Plateau, which is very good data source to investigate the characteristics of the

10 atmosphere-land interactions and turbulent exchanges of energy and mass in that region. However, after I read through the manuscript several times, I did not get more interesting findings except for knowing some numbers and magnitudes of several fluxes, which I think the study lacks significance and scientific contribution and therefore may need substantial work and more in-depth analyses and discussion.

15 **Author's Response**: **Accepted.** Following the general comments, we have made substantial revisions to make the manuscript more scientific-question-orientated. It mainly includes the following aspects.

(1) We enhanced the analysis and discussion on **the diurnal, seasonal, and inter-annual variability in surface energy fluxes**.

It is found that the energy balance components including net radiation, sensible heat, latent heat, and
20 soil heat fluxes all have similar diurnal courses, reaching their maximum values around noon and attaining their minimum values around early evening. Turbulent exchanges of energy fluxes are greatly influenced by the change of vegetation cover around the flux tower after 2014. The magnitudes of annual fluxes present remarkable trends with a large range. It is also found that the amplitude of LE is obviously larger than that of $H_s$ throughout the year, which indicates that the LE
25 plays a dominant role in surface heat exchange. Through comparative analysis, it is suggested that the characteristics of energy exchanges over the Tibetan Plateau, Yunnan-Guizhou Plateau, and their transitional zones are similar in the rainy season but obviously different in the dry season.

(2) We enhanced the analysis and discussion on **the diurnal, seasonal, and inter-annual variability in $CO_2$ flux**.

30 It is found that the study area acts as a weak carbon source at night because of plant and soil respiration, but acts as a carbon sink during the day thanks to $CO_2$ uptake from pant photosynthesis significantly exceeding $CO_2$ release though pant and soil respiration. The study area behaves as a carbon sink in all years. The annual total $F$c shows distinct jumps in magnitude, rapidly increasing

after 2014 and varying widely from $-966.9$ to $-75.6$ g C m$^{-2}$ yr$^{-1}$ with an average value of $-522.0$ g C m$^{-2}$. This is possibly because of the change of the underlying surface.

35

(3) We enhanced the discussion and explanation about **the relationships between turbulent fluxes of sensible heat, latent heat, and CO$_2$ and meteorological factors from half-hourly to yearly timescales**.

It is found that the wind speed is the most important controlling factor for sensible heat flux over all timescales, with their correlation coefficients becoming higher when the timescales becoming longer. It is likely caused by the local circulation existing in the Diancangshan Mountains and Erhai Lake, which may create more turbulent eddies and could promote the vertical exchange of heat, H$_2$O, and CO$_2$ between the atmosphere and cropland. The product of WS and VPD is the main environmental variable controlling LE. The effect of WS is not significant as VPD on LE when timescale is below monthly scale, whereas WS plays a decisive role in affecting LE on yearly scale. The relationship between CO$_2$ flux and meteorological factors are weak on half-hourly scale. The net radiation and soil temperature have the largest effects on $F$c from the daily to monthly scales. Similar to sensible and latent heat fluxes, wind speed is most significantly related to total CO$_2$ flux on yearly scale, which may be because of the climate effect of local circulation. Overall, the turbulent exchanges of energy, H$_2$O, and CO$_2$ fluxes are largely influenced by the WS, in particular on the yearly scale. This phenomenon is likely caused by the climate effect of local circulation in the complex terrain region.

As the authors described in the introduction, I agree it is very important to understand the atmosphere-land interactions in the transitional zone between the TP and Yunnan-Guizhou Plateau. However, the study site is an open and flat agricultural field and I wonder how well the flux measurements at the specific study site would be representative to the air-land interaction parameters in the transitional region, where more heterogeneity in land cover or topography may exist. The authors may need to stress more about the unique aspect of their study and reconsider: are the findings/results/data really relevant to what is currently described in the introduction and objectives?

**Author's Response: Yes.** The findings/results/data of this study station is really relevant to described in the introduction and objectives. Over the southeastern extension of the Tibetan Plateau (TP), that is, the transitional zone between the TP and Yunnan-Guizhou Plateau (YGP), there are many basins distributing in mountain valleys. The cropland is the main land-cover type and most residents live in these basins. Dali flux station, surrounded by open and flat agricultural field, is located in the basin between Diancangshan Mountains (DCM) and Yu'anshan Mountains. Erhai Lake, a highland shallow lake, is also located in here (Fig. 1b). This basin is similar to most basins and a typical representative in mountain environment over the southeastern extension of the TP. Furthermore, according to the footprint model analysis proposed by Kormann and Meixner (2001), it can be found that 95 % of source area contributing to the measured fluxes approximately came from 700 m in the southeast direction and

70     500 m in the northwest direction of the study site during 2007 to 2018 (Fig. 2). Although several roads, trees, buildings and irrigation channels were located in the source area, the underlying surface of source area was primarily covered by croplands, which satisfied the requirement of eddy covariance measurements, implying that the observed fluxes were reliable and mainly contributed by the croplands. Consequently, the flux measurements at this study site could be representative to the atmosphere-land

75     interaction parameters over the transitional region between the TP and YGP.

I also found that most of the results section is fully packed with numbers rather than result interpreting, for example, the authors spent one full page reporting all the meteorological data which are already clearly shown in Figure 2. I would suggest the authors could present the results by showing the key information in a concise way and including more in-depth analysis.

80 **Author's Response: Accepted.** We have made substantial revisions. The texts have been rewritten to show the findings clearly. The comparison between the findings of this study and some previous studies has been added. Rewritten contents have been added in Page 5 Line 145-159, Page 6-9 Line 173-276.

In addition, all the discussion is about the correlation analysis results, which should be moved to the results section. Therefore, I think the manuscript may need a better organization overall.

85 **Author's Response: Accepted.** The structure of the manuscript has been reorganized. The correlation analysis results have been moved to the results section. The section "Results" was changed to "Results and discussion". The subsection "Surface energy fluxes" was divided into three subsubsections ("Diurnal, seasonal and inter-annual variations in surface energy fluxes", "Controlling factors in $H_s$", and "Controlling factors in LE"). The subsection "$CO_2$ flux" was divided into two subsubsections

90     ("Diurnal, seasonal and inter-annual variations in $CO_2$ flux" and "Controlling factors in $CO_2$ flux").

I also noticed that as one of the key results, there is an obvious step change in LE, H, and Fc after 2014 (Figure 6 or 7). In particular, Hs becomes very noisy and doesn't show any obvious seasonal pattern after 2014, which also applies to the seasonal variability of Fc. This is very interesting, but I wonder whether it is really caused by the environmental controls or just a systematic error. Unfortunately, the

95     authors could not provide any explanation and discussion on this specific pattern of the inter-annual variability.

**Author's Response: Accepted.** There is an obvious step change for $H_s$ after 2013 and for LE and $F$c after 2014, which may mainly cause by the vegetation cover around the PBL flux tower. Relevant explanation and discussion on the inter-annual variability in $H_s$, LE, and $F$c have been added in Page 1

100     Line 17-19, Page 6-7 Line191-210, Page 9 Line 257-263, and Page 9 Line 285-286.

**Specific comments:**

Line 30-37: What kind of field campaigns are they? Are these studies all about turbulent exchanges and relevant to this study?

**Author's Response:** These field campaigns were carried out for mainly understanding atmospheric scientific questions over the TP and its surrounding areas, and the surface energy budget, turbulent exchanges, atmospheric boundary layer and etc. were taken as the important observational objectives and research tasks. A planetary boundary layer tower was established for continuously observing the micrometeorology and energy exchange at Dali, under the support of the JICA/Tibet Project. This study is relevant to these campaigns. The description about these campaigns has been rewritten in Page 2 Line 35-56.

Line 56: The study site is a cropland. However, from the google map (Figure1b), it seems the site is very close to the Erhai lake. Without a scale on the map, it is difficult to figure out whether there are any signals coming from the adjacent non-cropland land cover (e.g. the lake).

**Author's Response: Accepted.** Scale on the Google map (Figure 1b) and the average footprint source at study site (Figure 2) have been added. The study site is surrounded by open and flat farmland area (Fig. 1b-c) between the Diancangshan Mountains (DCM) and Erhai Lake. The distance from study site to DCM is approximately 4 km and to Erhai Lake is approximately 2 km. According to the footprint model analysis, the observed fluxes were reliable and mainly contributed by the croplands.

Line 90: Where are the global radiation and precipitation measurements coming from? Please describe in the methods section.

**Author's Response:** Global solar radiation was measured by the four-component net radiometer. Precipitation was measured with a tipping bucket rain gauge. Relevant describes have been added in Page 4 Line 113-116.

Line 90-106: This part of results is literally repeating the same information shown in Figure 2. Suggest rewriting the results in a concise way that shows the key information rather than reporting the numbers.

**Author's Response: Accepted.** This part of results has been rewritten in a concise way to show clearly the key information. Rewritten sentences have been added in Page 5 Line 145-159.

Line 111-113: Suggest moving this sentence to discussion.

**Author's Response:** In section 3, "Results" has been changed to "Results and discussion". Therefore, this sentence retains its original positon.

Line 117-118: Again, reporting numbers here. See my comment above.

**Author's Response: Accepted.** We have deleted the numbers.

Line 131-140: One could easily get this information from the figure.

**Author's Response: Accepted.** The part has been rewritten to show the findings clearly. The comparison between the findings of this study and some previous studies has been added. Rewritten contents have been added in Page 6 Line 173-187.

Line 170-185: These correlation analysis results are still "results", which should not be presented in the discussion. Suggest moving it the results section.

**Author's Response: Accepted.** We have moved these correlation analysis results to the section "Results".

Line 195: The inter-annual variability of annual carbon sink is really large, but I couldn't find there are any drivers or controls on this strong variability.

**Author's Response: Accepted.** The large inter-annual variability of annual carbon sink may be influenced by the changes of vegetation cover around the flux tower after 2014. Corresponding descriptions and discussion have been added in the sections "Abstract, Study site description, Results and discussion, Conclusions".

Line 198: Are there any physical or biophysical explanation on why wind speed affected the Hs?

**Author's Response: Yes.** It is well known that $H_s$ is primarily determined by the temperature difference between ground surface and atmosphere ($\Delta T$) as well as the turbulent exchange coefficient (Liu et al., 2012). There is a strong local circulation caused by the coupling effects of lake-land breeze, mountain-valley breeze, and gorge wind existing in the study area (Xu et al., 2011). This local circulation could modify wind speed and direction (Xu et al., 2020) and result in the air column to be vertically well mixed (Kutter et al., 2017). It may create more turbulent eddies and could promote the vertical exchange of heat, $H_2O$, and $CO_2$ between the atmosphere and cropland when the WS increases. Wind-induced mixing actively alters environmental variables, resulting in changes of turbulent exchange processes of heat and water vapour (Yusup and Liu, 2020).

Figure 1b: Please add the scale to the map and rotate the map with N facing upward.

**Author's Response: Accepted.** The scale to the map has been added. The map with N facing upward has been rotated. Additionally, three pictures of the Dali planetary boundary layer site have been added to show the changes of underlying surface.

Figure 3: How to distinguish the wind >= 28 m/s and 2-4 m/s, both are in yellow.

**Author's Response: Accepted.** The colour of the wind >= 28 m/s has been modified in wine.

Figure 5: Suggest adding the inter-annual variability of the monthly averaged diurnal patterns by showing the error bars.

**Author's Response: Accepted.** Figure 6 has been modified, showing the error bars.

Figure 6 and 7 basically show same information. Suggest deleting one.

**Author's Response: Accepted.** Figure 6 has been deleted, as suggested. Figure 7e has been changed in magenta in order to distinguish other variables.

**Technical corrections:**

Line 71: Hs instead of Ta?

**Author's Response: Thanks.** This mistake has been corrected.

Line 81: Suggest replacing "done" with "processed"

**Author's Response: Accepted.** The word has been changed, as suggested.

Line 82: Suggest deleting "used here"

**Author's Response: Accepted.** The words "used here" have been deleted.

Line 92: Round up 5066.1 to 5066. And please be careful about the significant digits throughout the manuscript.

**Author's Response: Accepted.** The numbers has been modified. In addition, we have careful checked the significant digits of the manuscript.

[Figure]

185 **Figure 1: Topographic map of the Tibetan Plateau (a), location of the study area (map from © Google Earth) (b), as well as pictures taken on 16 January 2009 (c), 26 September 2010 (d), 10 April 2016 (e) and 17 August 2016 (f) of the Dali flux station.**

[Figure]

**Figure 2: Average footprint source at Dali flux station from 2007 to 2018. The cumulative footprint contours were superimposed on the map from © Google Earth and the maximum footprint contour line is shown with 95 %.**

190

[Figure]

**Figure 4: Wind rose at a height of 10 m from 2007 to 2018.**

[Figure]

**Figure 6: Monthly average diurnal variations in energy balance components, including net radiation ($R_n$) (a), sensible heat flux ($H_s$)**
**(b), latent heat flux (LE) (c), soil heat flux ($G_s$) (d), and CO$_2$ flux ($Fc$) (e) from 2007 to 2018. The error bars indicate plus or minus**
**one standard deviation.**

[Figure]

Figure 7: Seasonal and inter-annual variations in monthly average energy balance components, including net radiation ($R_n$) (a), sensible heat flux ($H_s$) (b), latent heat flux (LE) (c), soil heat flux ($G_s$) (d), and CO$_2$ flux ($Fc$) (e) from 2007 to 2018.

**References**

Acharya, R. H., Sigdel, M., Ma, Y. M., and Wang, B. B.: Diurnal and seasonal variation of heat fluxes over an agricultural field in southeastern Nepal, Theor. Appl. Climatol., 137, 2949–2960, https://doi.org/10.1007/s00704-019-02790-3, 2019.

205 Alberto, R. C. M., Wassmann, R., Hirano, T., Miyata, A., Kumar, A., Padre, A., and Amante, M.: $CO_2$/heat fluxes in rice fields: comparative assessment of flooded and non-flooded fields in the Philippines, Agric. For. Meteorol., 149, 1737–1750, https://doi.org/10.1016/j.agrformet.2009.06.003, 2009.

Ajao, A. I., Jegede, O. O., and Ayoola, M. A.: Diurnal and seasonal variability of sensible and latent heat fluxes at an agricultural site in Ile-Ife, southwest Nigeria, Theor. Appl. Climatol., 139, 1237–1246, https://doi.org/10.1007/s00704-019-210 03043-z, 2020.

Baldocchi, D., and Meyers, T.: On using eco-physiological, micrometeorological and biogeochemical theory to evaluate carbon dioxide, water vapor and trace gas fluxes over vegetation: a perspective, Agric. For. Meteorol., 90, 1–25, https://doi.org/10.1016/S0168-1923(97)00072-5, 1998.

Bhattacharyya, P., Neogi, S., Roy, K. S. Dash, P. K., Tripathi, R., and Rao, K. S.: Net ecosystem $CO_2$ exchange and carbon 215 cycling in tropical lowland flooded rice ecosystem, Nutr. Cycl. Agroecosystems, 95, 133–144, https://doi.org/10.1007/s10705-013-9553-1, 2013.

Chen, L. X., Reiter, E. R., and Feng, Z. Q.: The atmospheric heat source over the Tibetan Plateau: May-August 1979, Mon. Weather Rev., 113, 1771–1790, https://doi.org/10.1175/1520-0493(1985)113<1771:TAHSOT>2.0.CO;2, 1985.

DeFries, R. S., Field, C. B., Fung, I., Justice, C. O., Los, S., Matson, P. A., Matthews, E., Mooney, H. A., Potter, C. S., 220 Prentice, K., Sellers, P. J., Townshend, J. R. G., Tucker, C. J., Ustin, S. L., and Vitousek, P. M.: Mapping the land surface for global atmosphere-biosphere models: toward continuous distributions of vegetation's functional properties, J. Geophys. Res., 100, 20867–20882, https://doi.org/10.1029/95JD01536, 1995.

Du, Q., Liu, H. Z., Xu, L. J., Liu, Y., and Wang, L.: The monsoon effect on energy and carbon exchange processes over a highland lake in the southwest of China, Atmos. Chem. Phys., 18, 15087–15104, https://doi.org/10.5194/acp-18-15087-2018, 225 2018a.

Du, Q., Liu, H. Z., Liu, Y., Wang, L., Xu, L. J., Sun, J. H., and Xu, A.L.: Factors controlling evaporation and the $CO_2$ flux over an open water lake in southwest of China on multiple temporal scales, Int. J. Climatol., 38, 1–17, https://doi.org/ 10.1002/joc.5692, 2018b.

Falge, E., Baldocchi, D., Olson, R. J., Anthoni, P., Aubinet, M., Bernhofer, C., Burba, G., Ceulemans, R., Clement, R., 230 Dolman, H., Granier, A., Gross, P., Grünwald, T., Hollinger, D., Jensen, N. O., Katul, G., Keronen, P., Kowalski, A., Ta Lai, C., Law, B. E., Meyers, T., Moncrieff, J., Moors, E., Munger, J. W., Pilegaard, K., Rannik, Ü., Rebmann, C., Suyker, A., Tenhunen, J., Tu, K., Verma, S., Vesala, T., Wilson, K., and Wofsy, S.: Gap filling strategies for defensible annual sums of net ecosystem exchange, Agric. For. Meteorol., 107, 43–69, https://doi.org/10.1016/S0168-1923(00)00225-2, 2001.

Kormann, R., and Meixner, F.X.: An analytical footprint model for non-neutral stratification, Bound.-Lay. Meteorol., 99, 235 207–224, https://doi.org/10.1023/a:1018991015119, 2001.

Kutter, E., Yi, C. X., Hendrey, G., Liu, H. P., Eaton, T., and Ni-Meister, W.: Recirculation over complex terrain, J. Geophys. Res. Atmos., 122, 6637–6651, https://doi.org/10.1002/2016JD026409, 2017.

Lei, H. M., Gong, T. T., Zhang, Y. C., and Yang, D. W.: Biological factors dominate the interannual variability of evapotranspiration in an irrigated cropland in the North China Plain, Agric. For. Meteorol., 250–251, 262–276, 240 https://doi.org/10.1016/j.agrformet.2018.01.007, 2018.

Li, G. P. (Eds.): Dynamic meteorology of the Tibetan Plateau, China Meteorological Press, Beijing, China, 97 pp., 2007.

Li, L., Vuichard, N., Viovy, N., Ciais, P., Wang, T., Ceschia, E., Jans, W., Wattenbach, M., Béziat, P., Gruenwald, T., Lehuger, S., and Bernhofer, C.: Importance of crop varieties and management practices: evaluation of a process-based model

for simulating $CO_2$ and $H_2O$ fluxes at five European maize (Zea mays L.) sites, Biogeosciences, 8, 1721–1736, https://doi.org/10.5194/bg-8-1721-2011, 2011.

Liu, H. P., Zhang, Q. Y., and Dowler, G.: Environmental controls on the surface energy budget over a large southern inland water in the United States: an analysis of one-year eddy covariance flux data, J. Hydrometeor., 13, 1893–1910, https://doi.org/10.1175/JHM-D-12-020.1, 2012.

Montagnani, L., Zanotelli, D., Tagliavini, M., and Tomelleri, E.: Timescale effects on the environmental control of carbon and water fluxes of an apple orchard, Ecol. Evol., 8, 416–434, https://doi.org/10.1002/ece3.3633, 2018.

Qian, Z. A., and Jiao, Y. J.: Advances and problems on Qinghai-Xizang Plateau meteorological research, Adv. Earth Sci., (in Chinese), 12, 207–216, 1997.

Reichstein, M., Falge, E., Baldocchi, D., Papale, D., Aubinet, M., Berbigier, P., Bernhofer, C., Buchmann, N., Gilmanov, T., Granier, A., Grünwald, T., Havránková, K., Ilvesniemi, H., Janous, D., Knohl, A., Laurila, T., Lohila, A., Loustau, D., Matteucci, G., Meyers, T., Miglietta, F., Ourcival, J. M., Pumpanen, J., Rambal, S., Rotenberg, E., Sanz, M., Tenhunen, J., Seufert, G., Vaccari, F., Vesala, T., Yakir, D., and Valentini, R.: On the separation of net ecosystem exchange into assimilation and ecosystem respiration: Review and improved algorithm, Glob. Change Biol., 11, 1424–1439, https://doi.org/10.1111/j.1365-2486.2005.001002.x, 2005.

Wang, Y., Zhou, L., Jia, Q. Y., and Oing, X. Y.: Direct and indirect effects of environmental factors on daily CO2 exchange in a rainfed maize cropland-A SEM analysis with 10 year observations, Field Crops Res., 242, 107591, https://doi.org/10.1016/j.fcr.2019.107591, 2019.

Wang, Y. J., Xu, X. D., Liu, H. Z., Li, Y. Q., Li, Y. H., Hu, Z. Y., Gao, X. Q., Mao, Y. M., Sun, J. H., Lenschow, D. H., Zhong, S. Y., Zhou, M. Y., Bian, X. D., and Zhao, P.: Analysis of land surface parameters and turbulence characteristics over the Tibetan Plateau and surrounding region, J. Geophys. Res., 121, 9540–9560, https://doi.org/10.1002/2016JD025401, 2016.

Wang, Y. W., Luo, W. J., Zeng, G. N., Peng, H. J., Cheng, A. Y., Zhang, L., Cai, X. L., Chen, J., Lyu, Y. N, Yang, H. L., and Wang, S. J.: Characteristics of carbon, water, and energy fluxes on abandoned farmland revealed by critical zone observation in the karst region of southwest China, Agric., Ecosyst. Environ., 292, 106821, https://doi.org/10.1016/j.agee.2020.106821, 2020.

Xin, Y. F., Chen, F., Zhao, P., Barlage, M., Blanken, P., Chen, Y. L., Chen, B., and Wang, Y. J.: Surface energy balance closure at ten sites over the Tibetan Plateau, Agric. For. Meteorol., 259, 317–328, https://doi.org/10.1016/j.agrformet.2018.05.007, 2018.

Xu, A. L., Zhao, X. H., Fu, Z. J., Liu, J. S., and Sun, J. H.: Comparison of meteorological elements over water and land surface in the Erhai Lake basin, Trans. Atmos. Sci., (in Chinese), 34, 225–231, 2011.

Xu, A. L., and Li, J.: An overview of the integrated meteorological observations in complex terrain region at Dali National Climate Observatory, China, Atmosphere, 11, 279, https://doi.org/10.3390/atmos11030279, 2020.

Xue, H. L., Li, J., Qian, T. T., and Gu, H. P.: A 100-m-scale modeling study of a gale event on the lee side of a long narrow mountain, J. Appl. Meteorol. Climatol, 59, 23–45, https://doi.org/10.1175/JAMc-D-190066.1, 2020.

Yusup, Y., and Liu, H. P.: Effects of atmospheric surface layer stability on turbulent fluxes of heat and water vapor across the water-atmosphere interface, J. Hydrometeor., 17, 2835–2851, https://doi.org/10.1175/JHM-D-16-0042.1, 2016.

Yusup, Y., and Liu, H. P.: Effects of persistent wind speeds on turbulent fluxes in the water-atmosphere interface, Theor. Appl. Climatol. 140, 313–325, https://doi.org/10.1007/s00704-019-03084-4, 2020.